# Color discrimination and gas chromatography-mass spectrometry fingerprint based on chemometrics analysis for the quality evaluation of Schizonepetae Spica

**Xindan Liu** , **Ying Zhang, Menghua Wu, Zhiguo Ma, Hui Cao** *

Research Center for Traditional Chinese Medicine of Lingnan, College of Pharmacy, Jinan University, Guangzhou, PR China

* kovhuicao@aliyun.com

**Data Availability Statement:** All relevant data are within the paper.

## Abstract

Schizonepetae Spica (SS), the dried spike of *Schizonepeta tenuifolia* Briq., is a traditional Chinese medicinal herb. According to the color of persistent calyx, SS is categorized into two classes: the yellowish-green-type and the brownish-type. Based on the chemometrics analysis of gas chromatography-mass spectrometry (GC-MS), a novel model of identifying and evaluating the quality of SS in different colors was constructed for the first time in this work. 20 batches SS samples of different colors were collected and used to extract essential oils. The average essential oils yield of SS in yellowish-green color was significantly higher than that of SS in brownish color from the same origin ($p<0.05$). The GC-MS fingerprints of 20 batches SS samples whose correlation coefficients were over 0.964 demonstrated SS samples were consistent to some extent in spite of slightly different chemical indexes. A total of 39 common volatiles compounds were identified. Hierarchical clustering analysis (HCA), principal component analysis (PCA) and partial least-squares discriminate analysis (PLS-DA) were developed to distinguish SS samples characterized by different colors. Consistent results were obtained to show that SS samples could be successfully grouped according to their color. Finally, 4,5,6,7-tetrahydro-3,6-dimethyl-benzofuran and pulegone were detected as the key variables for discriminating SS samples of different colors and for quality control. The obtained results proved that SS of good quality were often yellowish-green and those of poor quality were often brownish.

## Introduction

Color discrimination is an important aspect for the macroscopic identification of Chinese medicinal material (CMM).[1] Some ancient Chinese medicine literature, such as *Ben Cao Yuan Shi* (Origins of the Materia Medica), emphasize the importance of color as an identifier of high quality medicinal herbs. Modern pharmacologic studies have proven that colors of

**Funding:** This work was supported by the sixth batch of National Academic Experience Inheritance Project of Chinese Medicine Experts (No. 176-2017-XMZC-0166-01).

**Competing interests:** NO authors have competing interests.

some CMMs often have a large influence on the quality and quantity of the chemicals in medicinal herbs. For example, *Salvia miltiorrhiza* Bge. root (*dan seng*) in deep reddish-purple color is considered to be superior in quality, and modern experimental studies have demonstrated the deep reddish-purple color is indeed correlated with higher salvianolic acid B and tanshinones content.[2] *Aucklandia lappa* Decne. root (*mu xiang*) in blue-green color is thought to be superior, in yellowish-white color be mid-grade, and in black color be inferior in terms of quality.[3] In these cases, however, the correlation between color and chemical components is not well studied.

Schizonepetae Spica (SS, *jing jie sui*) is the dried spike of *Schizonepeta tenuifolia* Briq. (Chinese Pharmacopoeia, 2015 edition).[4] It was first recorded in *Shen Nong Ben Cao Jing* (Shen Nong's herbal classic), a book written 2000 years ago. SS is commonly used in traditional Chinese medicine (TCM) prescriptions to treat colds, allergic dermatitis, eczema and psoriasis.[5] Apart from a wide range of applications in traditional and modern medicine, it also has culinary applications of *S. tenuifolia* as ingredients in several food recipes, beneficial drinks and herbal tea.[6] Chemical studies revealed that SS contained volatile oils, flavonoids, organic acids, etc.[7–9] Research into pharmacological activities of the oils accumulated by SS possesses anti-inflammatory,[10] antioxidant,[11] analgesia,[12] antineoplastic[13, 14] and antiviral[15, 16] properties. Thus, the volatile components could be selected as marker compounds for quality evaluation of SS. One of the major pharmacological components in SS is pulegone, which is also quality control marker in the Chinese pharmacopoeia (2015 edition).[4] In China, SS is widely grown in Hebei province, Jiangsu province, Jiangxi province and many other regions. In the Chinese pharmacopoeia (2015 edition),[4] the departmental handbook "Dictionary of Chinese Traditional Medicine"[17] and "Dictionary of Chinese Medicine Processing",[18] SS is divided into two varieties according to the color of persistent calyx: the yellowish-green-type and the brownish-type.

Chromatographic fingerprinting is a unique pattern that indicates the presence of multiple biomarkers in a complex chemical system. The chromatographic fingerprint analysis could serve as a comprehensive approach for quality control of medicinal herbals and food products. [19] However, the minor differences between similar chromatograms generated by samples may not be readily detected. Multivariate statistical analyses, such as hierarchical clustering analysis (HCA), principal component analysis (PCA), and partial least-squares discriminate analysis (PLS-DA) have been proposed as proper tools to solve chromatographic problems and extract maximum useful information from the chromatographic fingerprinting.[20, 21]

In previous studies, gas chromatography-mass spectrometry (GC-MS) combined with chemometrics methods had been used to distinguish SS from different regions.[22, 23] However, few studies have done to analyze the correlation between the color and chemical composition in SS. Therefore, the investigation into the chemical difference of SS in different colors could be developed as an important approach to evaluate the quality of SS. In the present study, the volatile components of SS samples were acquired by GC-MS. Since SS samples in different colors may differ significantly, depending on a number of factors—such as plant cultivars, harvest time, processing, storage conditions and many other factors. The obtained information was then analyzed by multivariate methods including HCA, PCA and PLS-DA to find similarity among SS samples and to evaluate discriminating variables (biomarkers).

## Material and methods

### Materials and reagents

20 batches sample of medicinal herbals were purchased from various sources of China (Table 1), and their identities were confirmed as SS by Dr. Ying Zhang, Jinan University, PR

**Table 1. The details of the 20 batches Schizonepetae Spica samples and fingerprint similarities.**

| No. | Voucher specimens | R | G | B | Calyx color | Collection place | Date of collection | Similarities |
|---|---|---|---|---|---|---|---|---|
| 1 | YP132581301 | 90.72 | 74.52 | 40.40 | Brownish | Henan province | November 29, 2018 | 0.978 |
| 2 | YP132581401 | 81.27 | 62.55 | 31.23 | Brownish | Henan province | January 17, 2019 | 0.964 |
| 3 | YP132581501 | 92.32 | 98.24 | 45.06 | Yellowish-green | Guangdong province | January 8, 2019 | 0.996 |
| 4 | YP132581601 | 83.33 | 92.22 | 45.67 | Yellowish-green | Hubei province | January 9, 2019 | 0.996 |
| 5 | YP132581701 | 96.10 | 100.26 | 46.23 | Yellowish-green | Hubei province | January 10, 2019 | 0.987 |
| 6 | YP132581801 | 87.09 | 72.65 | 37.53 | Brownish | Shanxi province | January 9, 2019 | 0.988 |
| 7 | YP132581901 | 86.25 | 99.15 | 51.51 | Yellowish-green | Shanxi province | January 12, 2019 | 0.994 |
| 8 | YP132582001 | 81.02 | 92.38 | 45.57 | Yellowish-green | Anhui province | January 10, 2019 | 0.994 |
| 9 | YP132582101 | 81.51 | 68.26 | 38.24 | Brownish | Anhui province | January 19, 2019 | 0.999 |
| 10 | YP132580101 | 80.62 | 71.34 | 45.02 | Brownish | Hebei province | June 7, 2018 | 0.986 |
| 11 | YP132582201 | 80.59 | 95.29 | 39.18 | Yellowish-green | Jiangsu province | January 10, 2019 | 0.997 |
| 12 | YP132582301 | 97.19 | 105.54 | 52.24 | Yellowish-green | Hebei province | January 9, 2019 | 0.996 |
| 13 | YP132582401 | 85.38 | 98.34 | 43.10 | Yellowish-green | Hebei province | January 13, 2019 | 0.998 |
| 14 | YP132582501 | 86.35 | 97.09 | 40.83 | Yellowish-green | Zhejiang province | January 10, 2019 | 0.999 |
| 15 | YP132580701 | 82.43 | 68.02 | 40.78 | Brownish | Henan province | January 10, 2019 | 0.989 |
| 16 | YP132582601 | 91.48 | 101.2 | 53.97 | Yellowish-green | Jiangsu province | January 11, 2019 | 0.994 |
| 17 | YP132582701 | 80.12 | 74.90 | 41.14 | Brownish | Jiangsu province | January 11, 2019 | 0.989 |
| 18 | YP132582801 | 87.14 | 95.70 | 42.88 | Yellowish-green | Guangxi province | January 11, 2019 | 0.998 |
| 19 | YP132582901 | 92.76 | 101.42 | 48.61 | Yellowish-green | Jiangxi province | January 11, 2019 | 0.987 |
| 20 | YP132583001 | 84.17 | 70.81 | 44.45 | Brownish | Jiangxi province | January 12, 2019 | 0.992 |

R = red; G = green; B = blue.

China. Voucher specimens were deposited at the Research Center for Traditional Chinese Medicine of Lingnan, Jinan University.

All solvents used in the experiments were of analytical grade. Ethyl acetate was purchased from Aladdin (Aladdin, Shanghai, China). *n*-Alkane (purity > 97%) which used as an internal quality standard for GC-MS analysis was purchased from o2si (Charleston, SC, USA).

### Discrimination the color of Schizonepetae Spica calyx

Images were acquired using a digital camera (PowerShot G7 X Mark II, Canon Inc., Nagasaki, Japan) as shown in Fig 1. The red-green-blue (RGB) values were extracted from the images of samples using Photoshop CS6 (Adobe Systems Inc., USA) software (Table 1). The surface color of the Schizonepetae Spica calyx was selected to extract RGB values. And then the obtained RGB values were transformed into the CIE 1931XYZ color space, where the values can be normalized and plotted in a 2-dimensional CIE1931XYZ chromaticity diagram to identify the color of SS calyx as described by Wang et al.[24] According to the distribution of the SS samples in the CIE 1931XYZ chromaticity diagram (Fig 2), the yellowish-green-type consisted of samples 3, 4, 5, 7, 8, 11, 12, 13, 14, 16, 18 and 19; and the brownish-type contained samples 1, 2, 6, 9, 10, 15, 17 and 20 (Table 1).

### Essential oils extraction

Steam distillation method was chosen according to the Chinese Pharmacopoeia (2015 edition) for extraction of essential oils.[4] All samples were smashed and filtered through a 24 mesh sieve. Then the dried powder (50 g) was weighted and placed in a 1000 mL flask. 300 mL redistilled water was also placed in this 1000 mL flask. The essential oils were extracted by water

distillation for 4 h. At last, essential oils were separated from the water layer, and then oil layer was dried over anhydrous sodium sulfate. The extraction yield was calculated in a milliliter of essential oils per 100 g of SS. The anhydrous essential oils were stored in the dark glass vial at -20˚C. 50 mg of the essential oils sample were accurately weighted and transferred into a 5 mL volumetric flask, and make up to volume with ethyl acetate for GC-MS analysis.

## GC-MS analysis

Volatile compounds were analyzed on an Agilent 7890B GC system coupled to an Agilent 7000C GC/MS Triple Quad mass spectrometer (Agilent, Santa Clara, CA, USA). Initial chromatographic separations of 1 μL samples were on a 15 m × 250 μm i.d. × 0.25 μm film thickness HP-5 (Agilent) capillary column with a He flow rate of 1.0 mL/min and an injection port temperature of 250˚C with the split ratio of 1:10. The oven temperature ramp was 3 min at 50˚C, then 10˚C/min to 90˚C, where the temperature was held for 5 min, then ramped at 10˚C/min to 160˚C, where it was maintained for 10 min, then a 20˚C/min ramp to 260˚C, where the temperature was held for 3 min. The detector was operated at 70 eV ionization energy, and the $m/z$ values were recorded in range of 50–600 amu with a scan rate of 3.6 scan/s and a solvent delay of 3 min. Components were identified using the National Institute of Standards and Technology (NIST) 2.2L Mass Spectra Database containing about 189,000 compounds, as well as comparing with the literatures.[22, 25–27]

## Data analysis

The GC-MS fingerprint was performed by professional software Similarity Evaluation System for Chromatographic Fingerprint of Traditional Chinese Medicine (Version 2004 A)

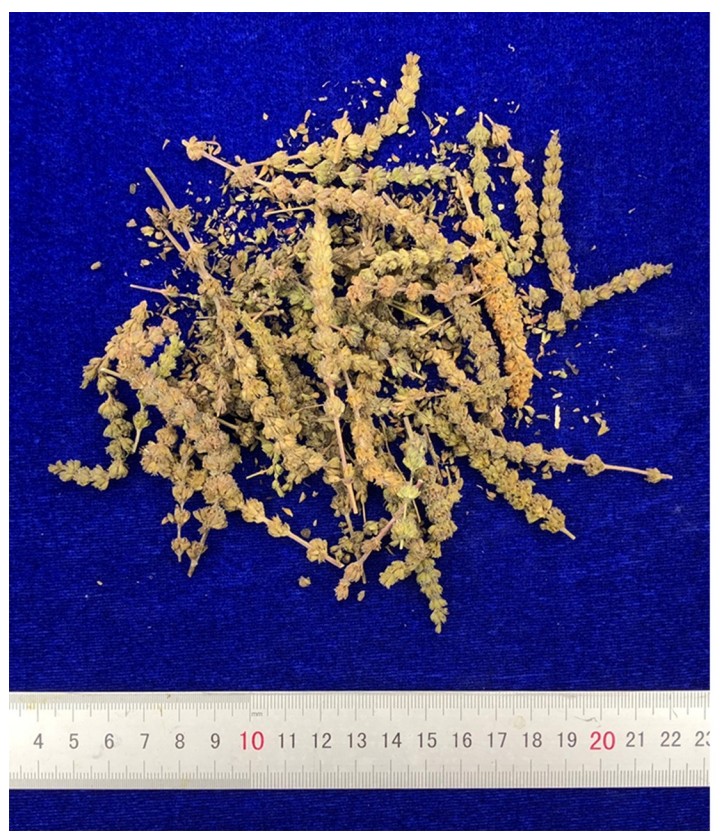 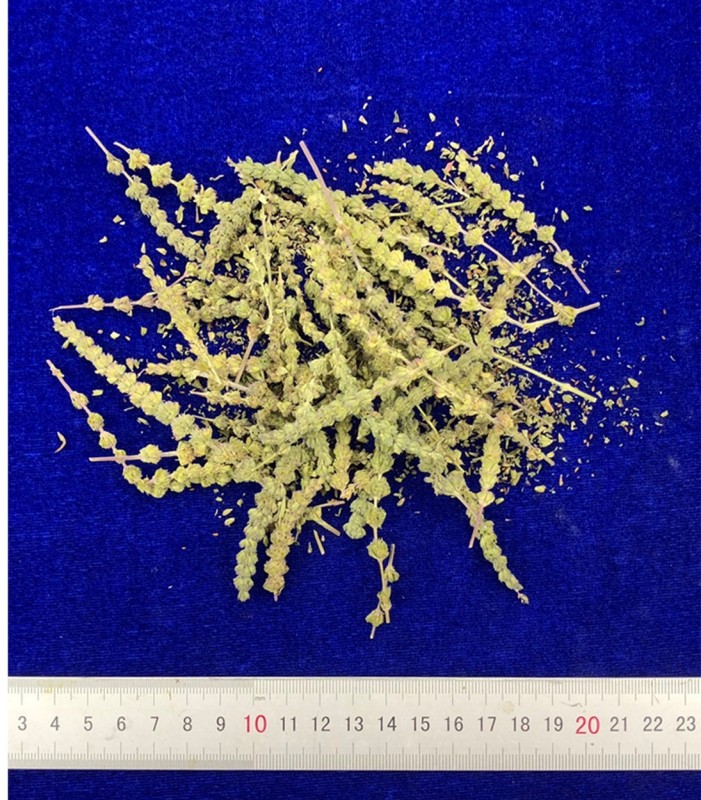

**Fig 1. Representative images of Schizonepetae Spica samples.**

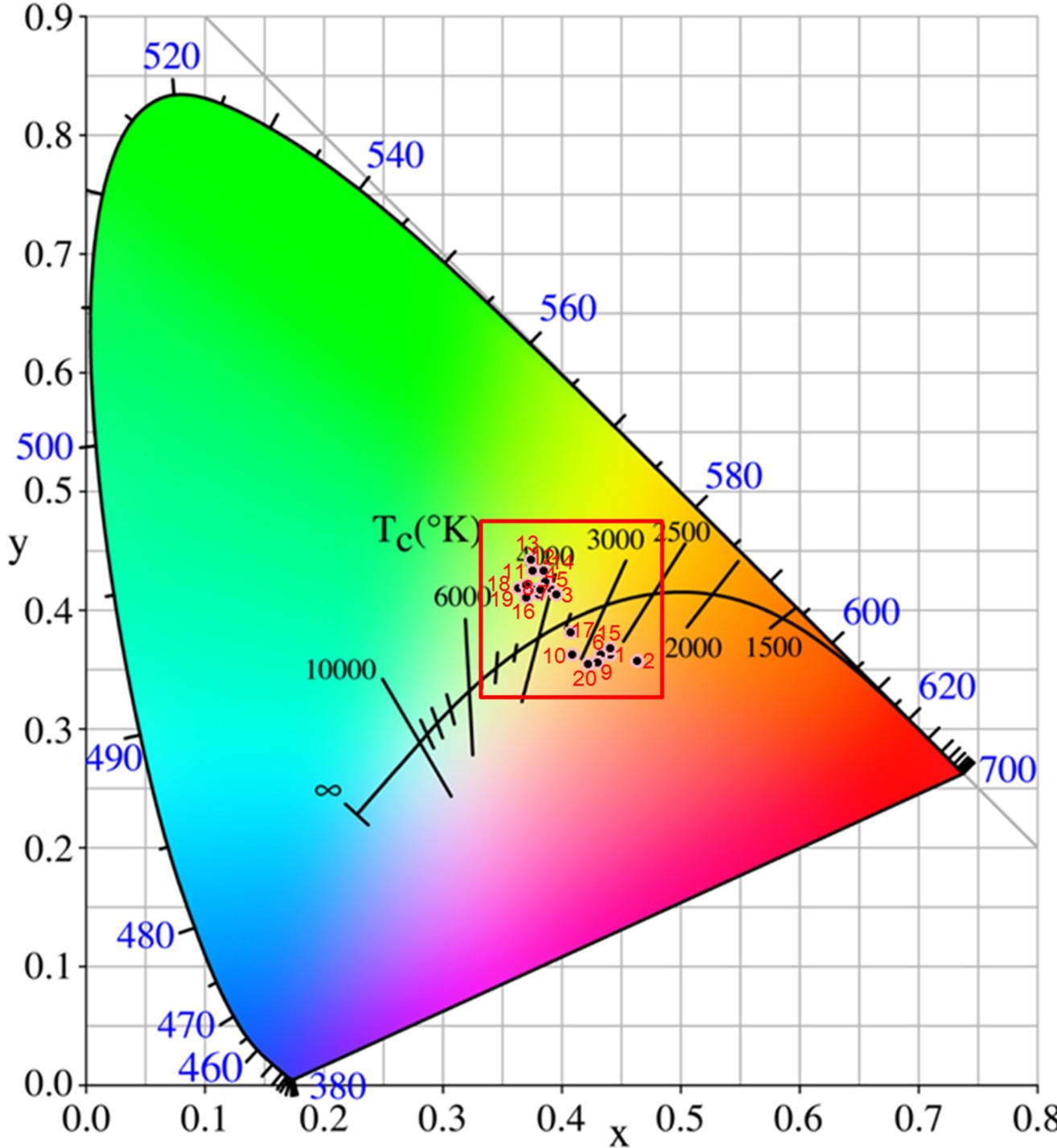

**Fig 2. The distribution of the Schizonepetae Spica samples in the CIE 1931XYZ chromaticity diagram.** The samples mainly distribute among the upper left region favoring yellowish-green; the samples distribute among the lower right tending to brownish.

composed by the Chinese Pharmacopoeia Committee. HCA was performed by SPSS version 20.0 software (SPSS, Chicago, USA) based on Ward's method and the squared Euclidean distance. PCA and PLS-DA were performed by SIMCA-P version 11.5 software (Umetrics,

Umea, Sweden). The loading plot from PLS-DA was employed to find chemical markers for discriminating the samples in different colors.[21, 28] Statistical significance was assessed by ANOVA test with GraphPad Prism v.5.0 software. *P*-values < 0.05 were considered to be statistically significant, and *p*-values < 0.01 were considered to be statistically highly significant.

## Results and discussion

### Yield of essential oils from Schizonepetae Spica of different colors

The essential oils from SS samples were extracted by hydrodistillation, and the distilled essential oils gave clear yellow wax oils in yields ranging from 0.47% to 1.65% mL/g (Fig 3). In this study, we acquired different colors of the SS, including the yellowish-green-type and the brownish-type. The yellowish-green color of SS corresponded to a relatively higher concentration of volatiles (~ 0.69%-1.65% mL/g), while SS in brownish color was correlated with relatively lower volatiles concentration (~ 0.47%-0.85% mL/g). Interestingly, significant differences in essential oils yield were detected between the yellowish-green-type and the brownish-type of SS from the same geographic origin (Fig 3), which indicated a relationship between the color of the herb and its volatiles concentration.

### Validation of methodology

The precision and repeatability of the method was determined by performing six injections of sample solution and six replicates of the same sample (No. 12), respectively (Table 1). The results showed that precision of relative retention times and relative peak areas of volatile constituents were found in the range of 0.00%-0.03% and 0.00%-3.12% of relative standard

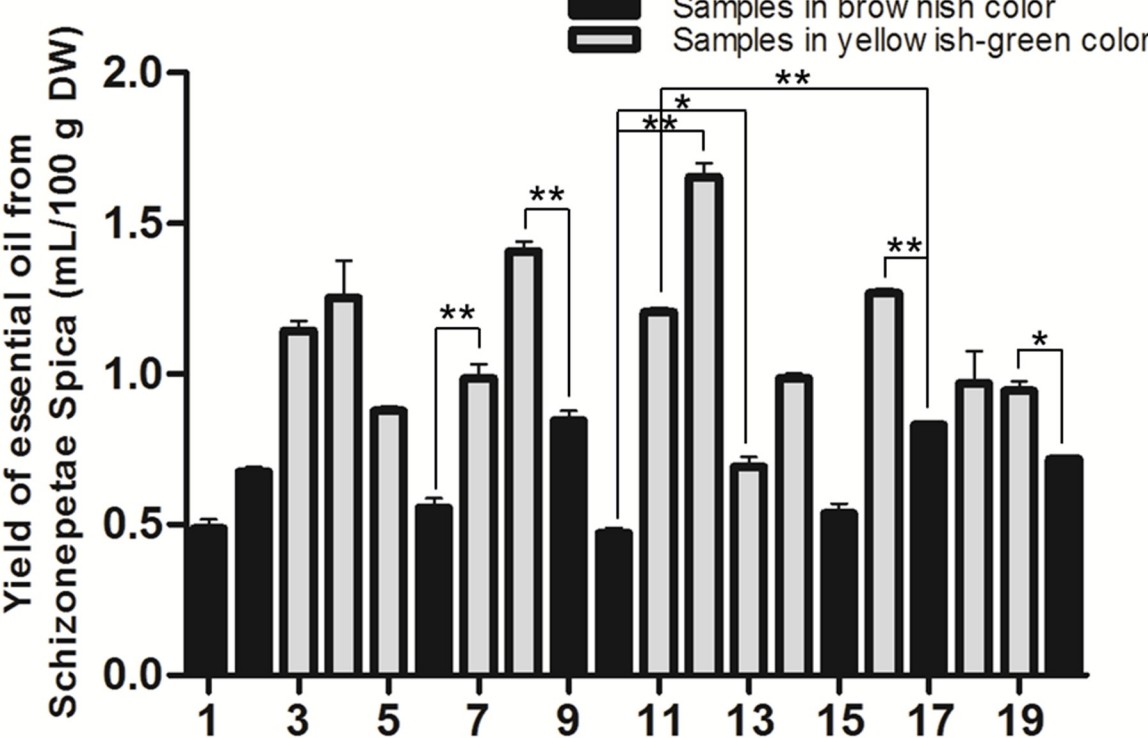

**Fig 3. Yield of essential oils from Schizonepetae Spica of different colors.** Data are presented as the mean ± SD (*n* = 2). *p<0.05, **p<0.01 *vs*. SS samples in brownish color from the same origin. DW = dry weight.

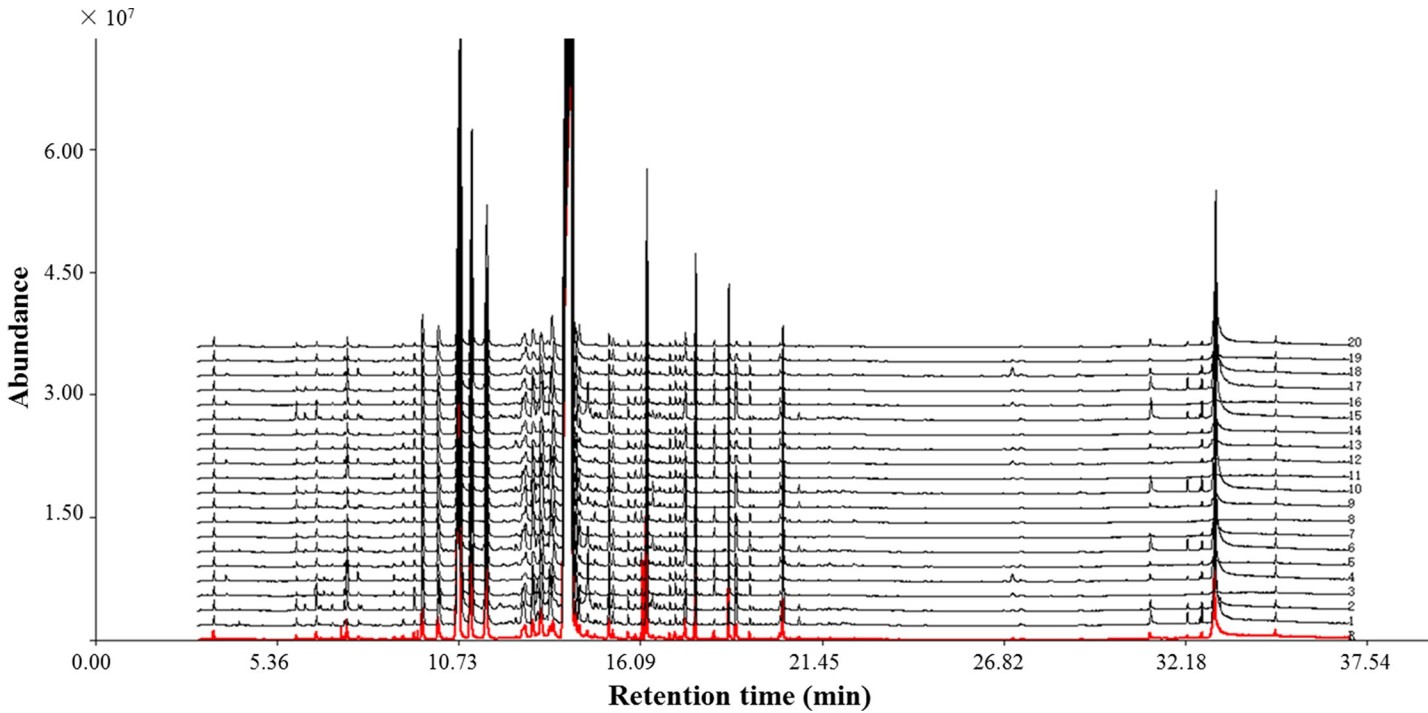

**Fig 4. GC-MS fingerprints of 20 batches of Schizonepetae Spica samples and reference chromatogram (R).**

deviation values (RSDs). The method repeatability of relative retention times and relative peak areas of volatile constituents were lower than 0.10% and 5.20% of RSDs, respectively. The method stability was determined at 0, 2, 4, 8, 12 and 24 h by using the same sample (No. 12). The RSDs of relative retention time and relative peak areas of volatile constituents were less than 0.08% and 5.50%, respectively. All the results indicated that the developed conditions for the GC-MS fingerprint analysis of SS were satisfactory.

### GC-MS fingerprint of essential oils from Schizonepetae Spica

The chromatograms of the SS samples (20 batches) were shown in Fig 4. The correlation coefficient of similarity between each chromatographic profile of SS and the reference chromatogram, the representative standard fingerprint/chromatogram for a group of chromatograms, were calculated (Table 1). The correlation coefficients of the 20 SS samples were more than 0.964, which were in agreement with previous studies.[29, 30] These results demonstrated that the chromatographic fingerprints of SS from different geographic origins were consistent to some extent in spite of slightly different chemical composition. Correlation coefficient of each chromatogram for 12 batches SS in yellowish-green color was found to be 0.987–0.999, while those of the 8 batches SS in brownish color were below 0.990 except No. 9 (0.999) and No. 20 (0.992) (Table 1). Slight difference in correlation coefficients demonstrated different internal qualities. Some samples characterized by the brownish color exhibited relatively high correlation coefficient could not be discriminated by their color using the correlation coefficients. Thus, because of the limitation of GC-MS fingerprint analysis for detecting minor differences in samples of different colors, pattern recognition analysis for quality control of SS was employed.

## Identification of volatile components from Schizonepetae Spica

A total of 39 common compounds were identified in SS samples of different colors, which amounted for about 89% of the total essential oils (Table 2). Among these identified volatiles, monoterpenoids especially pulegone and *L*-menthone were the major compounds in 20 batches SS samples, a result which was consistent with the findings of recent studies.[22, 25, 26] However, 39 compounds in SS of different colors showed differences in relative contents. The relative content of pulegone, caryophyllene and germacrene D were relatively higher in the yellowish-green-type samples than in the brownish-type samples; whereas another four main components *L*-menthone, 4,5,6,7-tetrahydro-3,6-dimethyl-benzofuran, caryophyllene oxide and (*Z,Z,Z*)-9,12,15-octadecatrienoic acid had a higher content in the brownish-type samples. The volatile components of SS have been recognized as the major constituents responsible for its biological effects. For example, pulegone, which is known for its pleasant odor, analgesia, anti-inflammatory and antiviral properties,[12, 31] is the chemical indicator in the Chinese Pharmacopoeia (2015 edition) of SS.[4] *L*-menthone also presents analgesia and antiviral effects.[31] Caryophyllene is a functional cannabinoid receptor (type 2) agonist. [32] 4,5,6,7-Tetrahydro-3,6-dimethyl-benzofuran is widely used as flavorings and fragrances. [33] In short, the differences among these variables may lead the difference in quality of SS samples characterized by different colors.

## Hierarchical clustering analysis

Based on the results of above GC-MS fingerprint analyses, 39 common volatile components in 20 batches SS samples were used to conduct HCA. The dendrogram showed that 20 batches SS samples could be divided into two groups (Fig 5): cluster I belonged to the brownish-type, while cluster II belonged to the yellowish-green-type except No. 9. Therefore, herbal color could be an important influencing factor for the content of volatile components in SS. The pattern distribution of No. 9 suggested that the plant cultivar, harvesting time, processing methods, storage condition or other poorly controlled aspects may influence sample quality and sample classification.[34, 35] Generally, samples in the same color were still clustered, which indicated that the internal quality of these samples was quite similar to each other.

## Principal component analysis

To further investigate the quality variation and differentiate the color of SS samples, PCA was performed based on the GC-MS data of 39 common compounds. The PCA ($R^2X$ = 0.885, $Q^2$ = 0.602) scores plot showed that the 20 batches samples were obviously separated from two groups according to the different calyx color of SS (Fig 6), which yielded the same result as HCA. The first two PCs explained 63.08% of data variance (PC1 = 41.53% and PC2 = 21.55%). Of which, "PC1" played a significant role in discriminating samples of different colors. It should be pointed out that the samples belonged to one color were acquired from different collecting sites (Table 1). Therefore, it is acceptable that these samples do not have same chromatographic profiles and so were assigned to corresponding classes with an unsupervised PCA.

## Partial least-squares discriminate analysis

Subsequently, a supervised PLS-DA was adopted to find out the specific variation between the yellowish-green-type and the brownish-type of SS. In this model, the parameters for the classification from SIMCA-P 11.5 software were $R^2Y$ = 0.905 and $Q^2$ = 0.808, which were stable and good to fitness and prediction, respectively. The score plot showed a clear categorization of

**Table 2. Thirty-nine common compounds and their relative contents of Schizonepetae Spica in different colors.**

| No. | Compound | Molecule formula | RI | CAS | Relative content (%) | |
|---|---|---|---|---|---|---|
| | | | | | Samples in yellowish-green color ($n = 12$) | Samples in brownish color ($n = 8$) |
| V1 | (R)-3-Methyl-cyclohexanone | $C_7H_{12}O$ | 953.74 | 13368-65-5 | 0.08 ± 0.02 | 0.14 ± 0.05 |
| V2 | 1-Octen-3-ol | $C_8H_{16}O$ | 983.39 | 3391-86-4 | 0.17 ± 0.03 | 0.17 ± 0.08 |
| V3 | 1,3,8-p-Menthatriene | $C_{10}H_{14}$ | 1005.59 | 18368-95-1 | 0.04 ± 0.01 | 0.04 ± 0.02 |
| V4 | E,E-2,6-Dimethyl-1,3,5,7-octatetraene | $C_{10}H_{14}$ | 1023.77 | 460-01-5 | 0.07 ± 0.02 | 0.07 ± 0.04 |
| V5 | D-Limonene | $C_{10}H_{16}$ | 1027.61 | 5989-27-5 | 0.41 ± 0.33 | 0.23 ± 0.12 |
| V6 | Benzeneacetaldehyde | $C_8H_8O$ | 1043.37 | 122-78-1 | 0.11 ± 0.05 | 0.09 ± 0.03 |
| V7 | 2-Ethenyl-1,4-dimethyl-benzene | $C_{10}H_{12}$ | 1089.74 | 2039-89-6 | 0.08 ± 0.03 | 0.05 ± 0.02 |
| V8 | Linalool | $C_{10}H_{18}O$ | 1100.42 | 78-70-6 | 0.10 ± 0.02 | 0.10 ± 0.02 |
| V9 | trans-1-Methyl-4-(1-methylethenyl)-2-cyclohexen-1-ol | $C_{10}H_{16}O$ | 1118.90 | 7212-40-0 | 0.76 ± 0.08 | 0.99 ± 0.19 |
| V10 | cis-p-Mentha-2,8-dien-1-ol | $C_{10}H_{16}O$ | 1133.26 | 3886-78-0 | 0.71 ± 0.08 | 0.87 ± 0.17 |
| V11 | L-Menthone | $C_{10}H_{18}O$ | 1151.53 | 14073-97-3 | 9.51 ± 3.72 | 16.95 ± 3.03 |
| V12 | 4,5,6,7-Tetrahydro-3,6-dimethyl-benzofuran | $C_{10}H_{14}O$ | 1160.88 | 494-90-6 | 1.35 ± 0.41 | 4.93 ± 1.34 |
| V13 | trans-5-Methyl-2-(1-methylethenyl)-cyclohexanone | $C_{10}H_{16}O$ | 1173.08 | 29606-79-9 | 2.63 ± 0.42 | 3.26 ± 0.63 |
| V14 | (-)-cis-Isopiperitenol | $C_{10}H_{16}O$ | 1202.38 | 96555-02-1 | 0.62 ± 0.18 | 0.94 ± 0.22 |
| V15 | 2-Allyl-4-methylphenol | $C_{10}H_{12}O$ | 1233.11 | 6628-06-4 | 0.70 ± 0.12 | 1.49 ± 0.44 |
| V16 | Pulegone | $C_{10}H_{16}O$ | 1253.74 | 89-82-7 | 67.51 ± 3.44 | 49.34 ± 5.22 |
| V17 | 3-Methyl-6-(1-methylethyl)-2-cyclohexen-1-one | $C_{10}H_{16}O$ | 1261.45 | 89-81-6 | 0.26 ± 0.16 | 0.49 ± 0.09 |
| V18 | 5-Formylmethyl-6-hydroxy-3,3-dimethyl-6-vinyl-bicyclo[3.2.0]heptan-2-one | $C_{13}H_{18}O_3$ | 1290.48 | 1000156-78-3 | 0.24 ± 0.03 | 0.57 ± 0.25 |
| V19 | (1S,3S,5S)-1-Isopropyl-4-methylenebicyclo[3.1.0]hexan-3-yl acetate | $C_{12}H_{18}O_2$ | 1294.44 | 139757-62-3 | 0.19 ± 0.02 | 0.19 ± 0.03 |
| V20 | Carveol | $C_{10}H_{16}O$ | 1314.62 | 99-48-9 | 0.11 ± 0.02 | 0.14 ± 0.04 |
| V21 | 2-Methyl-5-(1-methylethenyl)-2-cyclohexen-1-ol acetate | $C_{12}H_{18}O_2$ | 1335.28 | 97-42-7 | 0.12 ± 0.05 | 0.07 ± 0.02 |
| V22 | 3-Methyl-6-(1-methylethylidene)-2-cyclohexen-1-one | $C_{10}H_{14}O$ | 1344.05 | 491-09-8 | 3.44 ± 0.47 | 2.90 ± 0.54 |
| V23 | α-Copaene | $C_{15}H_{24}$ | 1378.56 | 1000360-33-0 | 0.13 ± 0.03 | 0.07 ± 0.02 |
| V24 | (-)-β-Bourbonene | $C_{15}H_{24}$ | 1386.74 | 5208-59-3 | 0.14 ± 0.04 | 0.11 ± 0.03 |
| V25 | 1-Ethenyl-1-methyl-2,4-bis(1-methylethenyl)-cyclohexane | $C_{15}H_{24}$ | 1393.37 | 515-13-9 | 0.11 ± 0.02 | 0.04 ± 0.01 |
| V26 | (Z)- 3-Methyl-2-(2-pentenyl)- 2-cyclopenten-1-one | $C_{11}H_{16}O$ | 1399.03 | 488-10-8 | 0.07 ± 0.01 | 0.07 ± 0.02 |
| V27 | 2-(2-Butenyl)-4-hydroxy-3-methyl-2-cyclopenten-1-one | $C_{10}H_{14}O_2$ | 1401.35 | 17190-74-8 | 0.29 ± 0.13 | 0.50 ± 0.18 |
| V28 | Caryophyllene | $C_{15}H_{24}$ | 1422.16 | 87-44-5 | 1.84 ± 0.34 | 0.54 ± 0.40 |
| V29 | Humulene | $C_{15}H_{24}$ | 1458.11 | 6753-98-6 | 0.21 ± 0.05 | 0.06 ± 0.04 |
| V30 | Germacrene D | $C_{15}H_{24}$ | 1485.14 | 23986-74-5 | 1.21 ± 0.26 | 0.28 ± 0.25 |
| V31 | 1,2,3,5,6,8a-Hexahydro-4,7-dimethyl-1-(1-methylethyl)-naphthalene | $C_{15}H_{24}$ | 1524.78 | 483-76-1 | 0.15 ± 0.03 | 0.09 ± 0.06 |
| V32 | (-)-Spathulenol | $C_{15}H_{24}O$ | 1581.27 | 77171-55-2 | 0.13 ± 0.04 | 0.14 ± 0.05 |
| V33 | Caryophyllene oxide | $C_{15}H_{24}O$ | 1586.74 | 1139-30-6 | 0.42 ± 0.19 | 1.28 ± 0.61 |
| V34 | (1R,3E,7E,11R)-1,5,5,8-Tetramethyl-12-oxabicyclo[9.1.0]dodeca-3,7-diene | $C_{15}H_{24}O$ | 1612.71 | 19888-34-7 | 0.03 ± 0.02 | 0.09 ± 0.05 |
| V35 | (1R,7S,E)-7-Isopropyl-4,10-dimethylenecyclodec-5-enol | $C_{15}H_{24}O$ | 1690.12 | 81968-62-9 | 0.05 ± 0.02 | 0.02 ± 0.01 |
| V36 | 6,10,14-Trimethyl-2-pentadecanone | $C_{18}H_{36}O$ | 1842.16 | 502-69-2 | 0.06 ± 0.02 | 0.04 ± 0.02 |
| V37 | (Z,Z,Z)-9,12,15-Octadecatrienoic acid, methyl ester | $C_{19}H_{32}O_2$ | 2100.00 | 301-00-8 | 0.06 ± 0.03 | 0.12 ± 0.06 |
| V38 | (Z,Z,Z)-9,12,15-Octadecatrienoic acid | $C_{18}H_{30}O_2$ | 2149.50 | 463-40-1 | 1.30 ± 1.52 | 4.76 ± 1.96 |
| V39 | 2,2'-Methylenebis[6-(1,1-dimethylethyl)-4-methyl-phenol | $C_{23}H_{32}O_2$ | 2429.11 | 119-47-1 | 0.11 ± 0.02 | 0.09 ± 0.01 |

Relative content (%) in the last two columns represents the mean ± SD. RI, retention index. CAS, Chemical Abstracts Service.

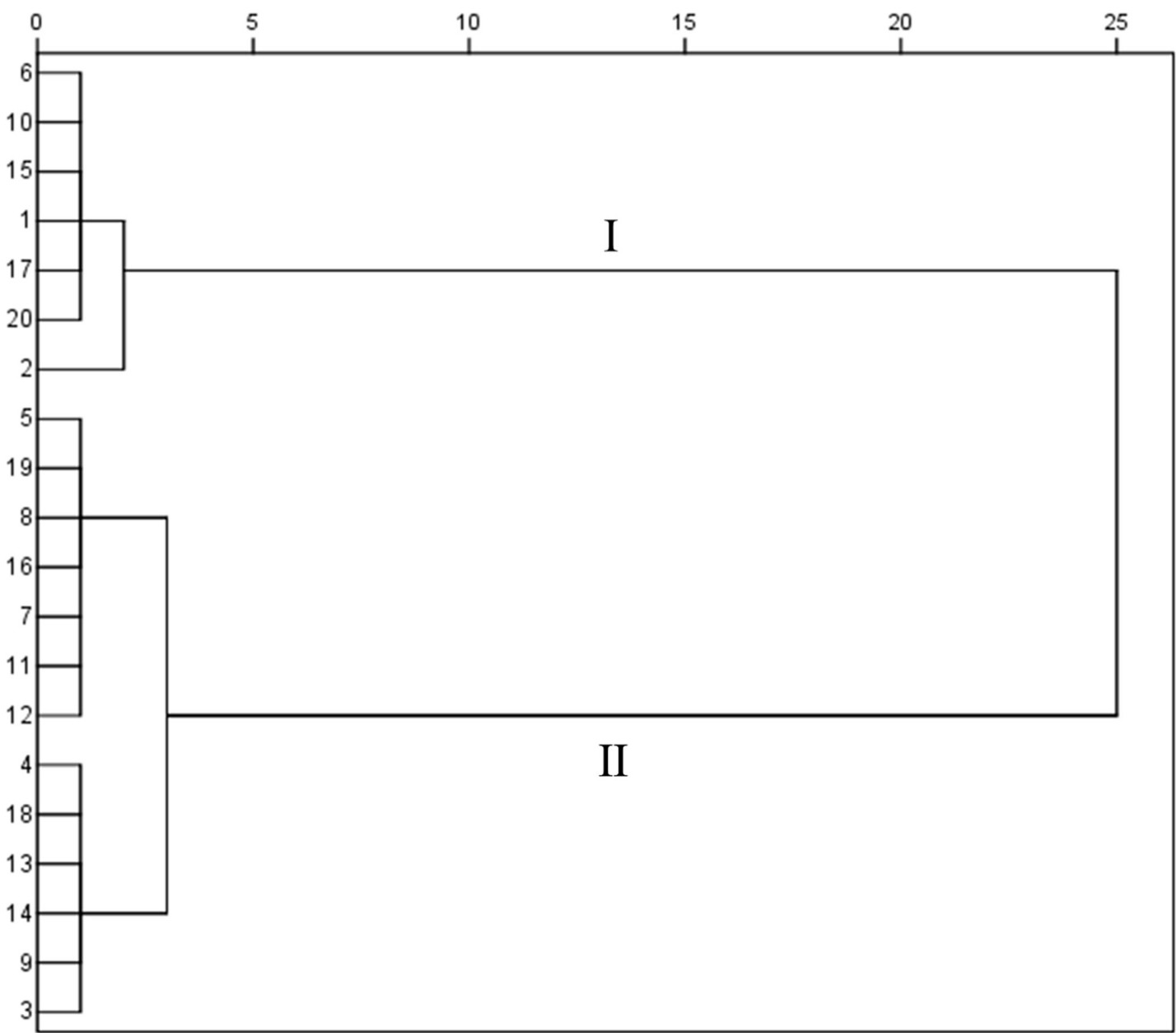

**Fig 5. Dendrogram obtained from hierarchical clustering analysis of 39 common volatile components of 20 batches Schizonepetae Spica samples in different colors.**

two classes: the yellowish-green-type and the brownish-type (Fig 7A). The results were in very good agreement with those obtained from HCA and PCA scores plot. To further validate the quality of the PLS-DA tests, random permutation class membership and the performance of 200 iterations were conducted. The $R^2$ and $Q^2$ intercept values were 0.352 and -0.23 and thus showed a low risk of over fitting and reliable (Fig 7B). Based on the PLS-DA, a loading plot was used to select the significant components that were differentially produced in the yellowish-green-type and the brownish-type of SS. The discriminant variables whose variable importance plot (VIP) are larger than 1.25 were selected as the significant different fragments. It could be seen from Fig 7C that V12 (4,5,6,7-tetrahydro-3,6-dimethyl-benzofuran), V16

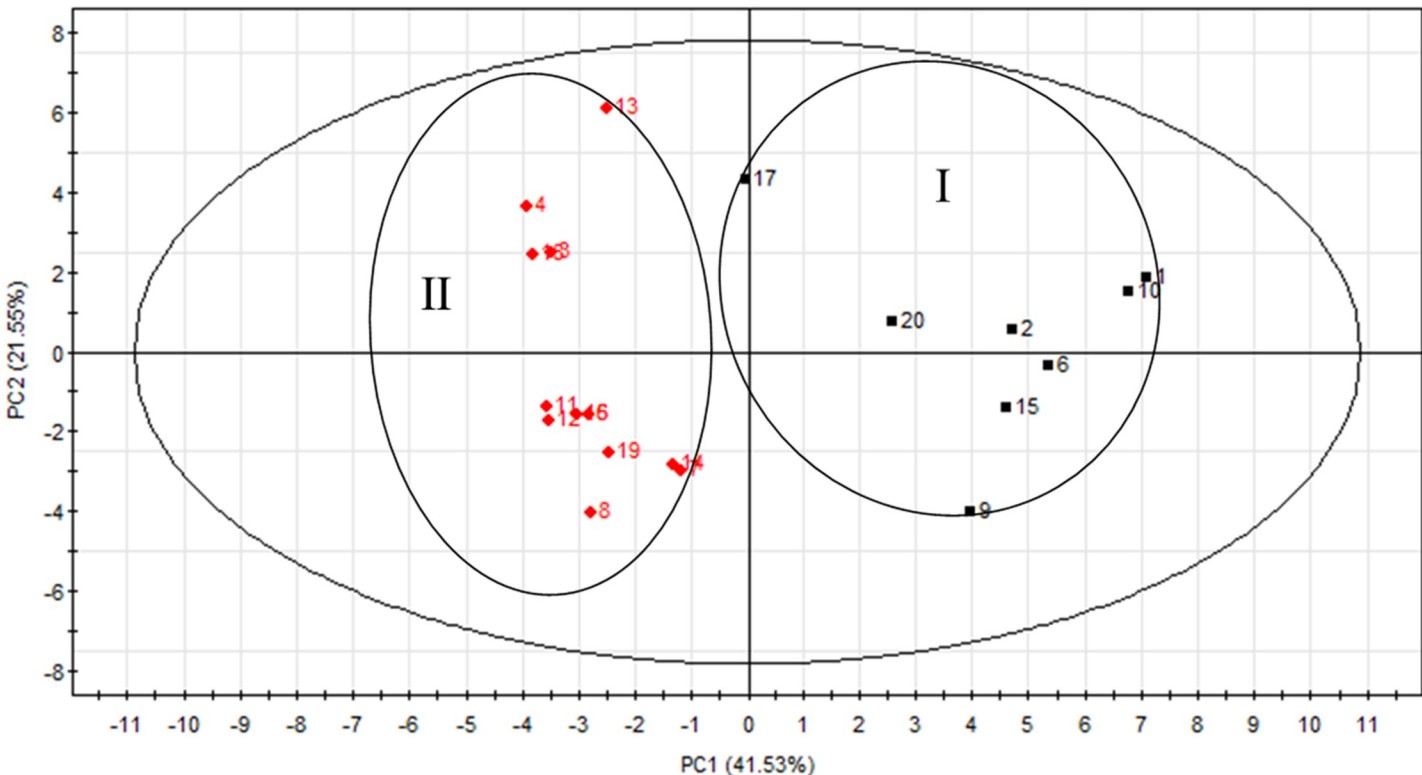

**Fig 6. PCA scores plot for volatile oils in Schizonepetae Spica by GC-MS.**

(pulegone), V30 (germacrene D), V28 (caryophyllene), V29 (humulene), V25 (1-ethenyl-1-methyl-2,4-bis(1-methylethyl)-cyclohexane), V15 (2-allyl-4-methylphenol), V38 ((Z,Z,Z)-9,12,15-octadecatrienoic acid), V13 (*trans*-5-methyl-2-(1-methylethenyl)-cyclohexanone) and V11 (*L*-menthone) might be the discriminant compounds in distinguishing the two color varieties of SS samples (Table 1). The variable importance plot for PC1 and PC2 indicated that V12 (4,5,6,7-tetrahydro-3,6-dimethyl-benzofuran) and V16 (pulegone) may have greater influence on the discrimination between the yellowish-green-type and the brownish-type of SS samples. The relative content of 4,5,6,7-tetrahydro-3,6-dimethyl-benzofuran was higher in the brownish-type samples than in the yellowish-green-type samples, while the relative content of pulegone was higher in the yellowish-green-type than in the brownish-type samples (Table 2). Interestingly, these two constituents had been demonstrated with good biological activities,[4, 12, 31, 33] and pulegone was also quality marker in the Chinese Pharmacopoeia (2015 edition).[4] Therefore, the relative content of these two compounds can be applied as marker components for quality evaluation of SS samples in different colors and two chemotypes of SS were proposed.

According to previous studies,[26, 36] 4,5,6,7-tetrahydro-3,6-dimethyl-benzofuran is a hepatotoxin and a major oxidative metabolite of pulegone, and cytochrome P450 enzyme (CYP) involved in the biosynthesis pathway had been identified in mint species. Building on the extensive genetic and biochemical knowledge accumulated for the genus *Mentha*, as calyx changed from yellowish-green to brownish, the plausible biotransformation in SS was hypothesized as shown in Fig 8.[37] However, evidence has been proved that pulegone, 4,5,6,7-tetrahydro-3,6-dimethyl-benzofuran and their oxidative product *p*-cresol could cause severe hepatotoxicity.[26] Thus, the pharmacological function should be further investigated to ensure the therapeutic effect and safety of SS in different colors.

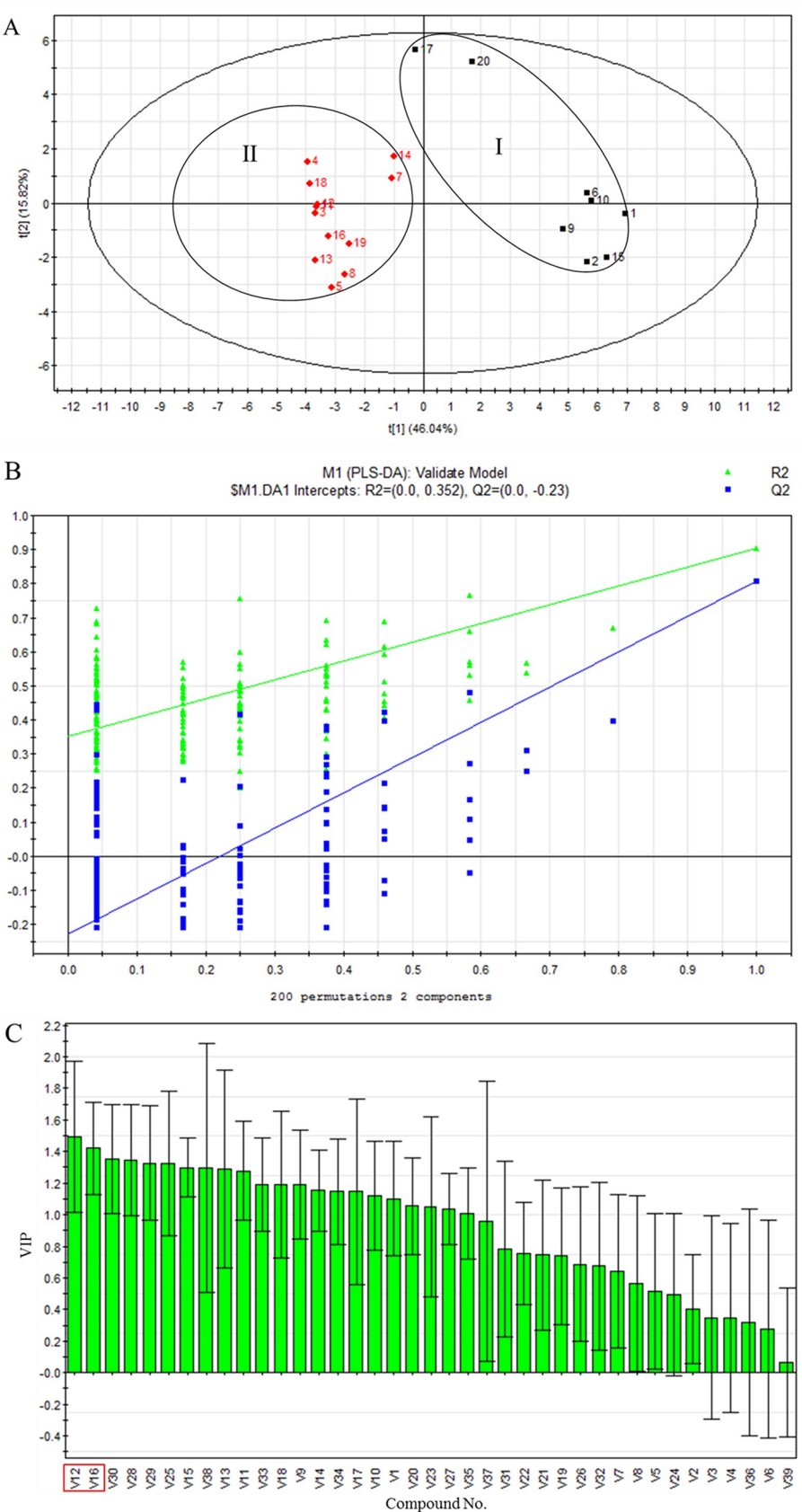

**Fig 7.** PLS-DA scores plot (A), $R^2$ and $Q^2$ intercept values from 200 iterations (B), and variable importance plot (C) of identified compounds in volatile oils of 20 batches Schizonepetae Spica samples in different colors.

**Fig 8. Plausible biotransformation pathway in Schizonepetae Spica when calyx changed from yellowish-green to brownish.** CYP: cytochrome P450 enzyme.

## Conclusion

This work reported for the first time applying the fingerprint analysis technology combined with chemometrics methods to characterize, discriminate and evaluate the quality of SS samples in different colors. The GC-MS fingerprints of 20 batches SS samples had relatively high similarity values ($\geq 0.964$) and 39 common components were identified. Accordingly, by using HCA, PCA and PLS-DA for analyzing the obtained results, the SS samples were categorized into two classes: the yellowish-green-type and the brownish-type. The amount of 4,5,6,7-tetrahydro-3,6-dimethyl-benzofuran and pulegone were identified as the major biomarkers of the SS quality which directly depend on the calyx color of samples. Our results demonstrated that samples of SS that had yellowish-green calyx was thought to be superior while those had brownish calyx were considered to be inferior in terms of quality. In summary, marker variables and correlation coefficients of GC-MS fingerprints between the yellowish-green-type and the brownish-type provided new clues to evaluate the quality of SS, as well as provided useful references for the standardization of color-based quality control for medicinal herbals, foods and other products.

## Author Contributions

**Conceptualization:** Hui Cao.

**Data curation:** Xindan Liu.

**Formal analysis:** Xindan Liu.

**Funding acquisition:** Hui Cao.

**Methodology:** Menghua Wu, Zhiguo Ma.

**Project administration:** Hui Cao.

**Resources:** Ying Zhang.

**Writing – original draft:** Xindan Liu, Hui Cao.

**Writing – review & editing:** Xindan Liu, Ying Zhang, Menghua Wu, Zhiguo Ma, Hui Cao.

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
