## [Decision Letter · Decision Letter 0]

27 Nov 2019

PONE-D-19-27294

Color discrimination and gas chromatography-mass spectrometry fingerprint based on chemometrics analysis for the quality evaluation of Schizonepetae Spica

PLOS ONE

Dear Prof. Cao,

Thank you for submitting your manuscript to PLOS ONE. After careful consideration, we feel that it has merit but does not fully meet PLOS ONE’s publication criteria as it currently stands. Therefore, we invite you to submit a revised version of the manuscript that addresses the points raised during the review process.

Please see Academic Editor and Reviewer #1 comments below.

We would appreciate receiving your revised manuscript by Jan 10 2020 11:59PM. To enhance the reproducibility of your results, we recommend that if applicable you deposit your laboratory protocols in protocols.io, where a protocol can be assigned its own identifier (DOI) such that it can be cited independently in the future. For instructions see: http://journals.plos.org/plosone/s/submission-guidelines#loc-laboratory-protocols

We look forward to receiving your revised manuscript.

Kind regards,

Joseph Banoub, Ph,D., D. Sc.

Academic Editor

PLOS ONE

Journal Requirements:

Academic Editor comments to authors:

Color discrimination is an important aspect for the macroscopic identification of Chinese medicinal material (CMM). In this manuscript, the authors explain that the color of the persistent calyx of Schizonepetae Spica (SS) herb can be categorized into two classes: the yellowish-green-type and the brownish-type.

The authors described a novel approach based on gas chromatography-mass spectrometry (GC-MS) to identify and evaluate the quality of SS of different colors. The authors used 20 different colors batches of SS and extracted their essential oils. The authors found that the average essential oils yield of yellowish-green color SS was significantly higher than that of SS having the brownish color.

The GC-EI-MS fingerprints of 20 batches SS samples indicated the presence of a total of 39 identified common volatiles compounds.

The authors used hierarchical clustering analysis (HCA), principal component analysis (PCA) and partial least-squares discriminate analysis (PLS-DA) to distinguish SS samples characterized by different colors.

The authors showed that 4,5,6,7-tetrahydro-3,6-dimethyl-benzofuran and pulegone were detected as the key variables for discriminating SS samples of different colors and for quality control.

In summary, this manuscript is well written and presents a straightforward method for detecting the marker variables and correlation coefficients of GC-MS fingerprints between the yellowish-green-type and the brownish-type provided new clues to evaluate the quality of SS.

1.In page 3 of your pdf, lines 43-45, you have written the following:

Color discrimination is an important aspect for the macroscopic identification of 44 Chinese medicinal material (CMM) and for the evaluation of quality in traditional 45 experiences.[1]

QUERY: Please re-write as it makes no sense.

2. In page 6-7, Table 1. The details of the 20 batches Schizonepetae Spica samples and fingerprint 109 similarities.

QUERY: Can you please explain how did you obtain the values of R, G and B. this is not clear at all. Perhaps this table should be inserted after you discuss the Discrimination the color of Schizonepetae Spica calyx section.

Reviewers' comments:

Reviewer's Responses to Questions

**Comments to the Author**

1. Is the manuscript technically sound, and do the data support the conclusions?

Reviewer #1: Yes

2. Has the statistical analysis been performed appropriately and rigorously? 

Reviewer #1: I Don't Know

3. Have the authors made all data underlying the findings in their manuscript fully available?

Reviewer #1: Yes

4. Is the manuscript presented in an intelligible fashion and written in standard English?

Reviewer #1: Yes

5. Review Comments to the Author

Reviewer #1: Manuscript: PONE-D-19-2729

In this Manuscript, the authors used GC-MS to differentiate between the two different classes of Schizonepetae Spica (SS), a traditional Chinese medicinal herb. The SS samples can be classified according to the color of the calyx in to the yellowish-green-type and the brownish-type.

The authors performed GC-MS analysis on the essential oils extracted from 20 different SS samples. The resulting 20 GC-MS chromatograms are almost identical and allowed the identification of 39 common volatile compounds in all samples. The relative content of pulegone, caryophyllene and germacrene D were relatively higher in the yellowish-green-type samples than in the brownish-type samples; whereas another four main components L-menthone, 4,5,6,7-tetrahydro-3,6-dimethyl-benzofuran, caryophyllene oxide and (Z,Z,Z)-9,12,15-octadecatrienoic acid had a higher content in the brownish-type samples.

Due to the limitation of GC-MS fingerprint analysis for detecting minor differences in samples of different colors, some statistical analysis (HCA, PCA and PLS-DA) was used to find chemical markers for the discrimination between the samples of different colors. Based on these statistical analyses, the SS samples were categorized into two classes: the yellowish-green-type and the brownish-type. The amount of 4,5,6,7-tetrahydro-3,6-dimethyl-benzofuran and pulegone were identified as the major biomarkers of the SS quality which directly depend on the calyx color of samples. The relative content of 4,5,6,7-tetrahydro-3,6-dimethyl-benzofuran was higher in the brownish-type samples than in the yellowish-green-type samples, while the relative content of pulegone was higher in the yellowish-green-type than in the brownish-type samples. Therefore, the relative content of these two compounds can be applied as marker components for quality evaluation of SS samples in different colors.

Minor Comments

1- In the conclusion section, the authors mentioned the following “ Our results demonstrated that samples of SS that had yellowish-green calyx was thought to be superior while those had brownish calyx were considered to be inferior in terms of quality” Please explain this conclusion in more details as it is not clear in the manuscript.

2- I recommend to demonstrate this section Identification of volatile components from Schizonepetae Spica before the section of Hierarchical clustering analysis

3- In table 2, Molecule structure should be Molecular Formula

6. PLOS authors have the option to publish the peer review history of their article (what does this mean?). If published, this will include your full peer review and any attached files.

Reviewer #1: No

---

## [Author Response · Author response to Decision Letter 0]

12 Dec 2019

Reviewer #1

1. The yellowish-green color of SS corresponded to a relatively higher concentration of volatiles (~ 0.69%-1.65% mL/g), while SS in brownish color was correlated with relatively lower volatiles concentration (~ 0.47%-0.85% mL/g) (in section Yield of essential oils from Schizonepetae Spica of different colors). In addition, pulegone, which is known for its pleasant odor, analgesia, anti-inflammatory and antiviral properties, is the chemical indicator in the Chinese Pharmacopoeia (2015 edition) of SS (in section Identification of volatile components from Schizonepetae Spica and section Partial least-squares discriminate analysis). In our manuscript, we demonstrated that the relative content of 4,5,6,7-tetrahydro-3,6-dimethyl-benzofuran and pulegone could be applied as marker components for quality evaluation of SS samples in different colors. And the relative content of pulegone was higher in the yellowish-green-type than in the brownish-type samples (in section Partial least-squares discriminate analysis). Therefore, we can draw a conclusion that SS of good quality were often yellowish-green and those of poor quality were often brownish.

2. Thank you for your kindly recommendation. The section Identification of volatile components from Schizonepetae Spica has been inserted before the section of Hierarchical clustering analysis in the manuscript.

3. In table 2, “Molecule structure” has been revised as “Molecular Formula”.

Academic editor

1. The sentence “color discrimination is an important aspect for the macroscopic identification of Chinese medicinal material (CMM) and for the evaluation of quality in traditional experiences” has been revised as “color discrimination is an important aspect for the macroscopic identification of Chinese medicinal material (CMM)” as shown in section Introduction.

2. The red-green-blue (RGB) values were extracted from the images of samples using Photoshop CS6 (Adobe Systems Inc., USA) software (Table 1). The surface color of the Schizonepetae Spica calyx was selected to extract RGB values. Thank you for your kindly recommendation. Table 1 has been inserted after the Discrimination the color of Schizonepetae Spica calyx section.

---

## [Editor Report · Decision Letter 1]

16 Dec 2019

Color discrimination and gas chromatography-mass spectrometry fingerprint based on chemometrics analysis for the quality evaluation of Schizonepetae Spica

PONE-D-19-27294R1

Dear Dr. Cao,

We are pleased to inform you that your manuscript has been judged scientifically suitable for publication and will be formally accepted for publication once it complies with all outstanding technical requirements.

With kind regards,

Joseph Banoub, Ph,D., D. Sc.

Academic Editor

PLOS ONE

Additional Editor Comments (optional):

I am sending you this letter to confirm that your manuscript is now acceptable for publication
---

## [Editor Report · Acceptance letter]

20 Dec 2019

PONE-D-19-27294R1 

Color discrimination and gas chromatography-mass spectrometry fingerprint based on chemometrics analysis for the quality evaluation of Schizonepetae Spica 

Dear Dr. Cao:

I am pleased to inform you that your manuscript has been deemed suitable for publication in PLOS ONE. Congratulations! Your manuscript is now with our production department. 

With kind regards,

on behalf of

Dr. Joseph Banoub 

Academic Editor

PLOS ONE